# Development of a modular stress management platform (Performance Edge VR) and a pilot efficacy trial of a bio-feedback enhanced training module for controlled breathing

Murielle G. Kluge[1,2☯], Steven Maltby[1,2☯], Nicole Walker[3], Neanne Bennett[4], Eugene Aidman[2,5,6], Eugene Nalivaiko[1,2‡], Frederick Rohan Walker[1,2‡]*

1 Centre for Advanced Training Systems, Faculty of Health & Medicine, The University of Newcastle, Callaghan, NSW, Australia, 2 School of Biomedical Sciences & Pharmacy, Faculty of Health & Medicine, The University of Newcastle, Callaghan, NSW, Australia, 3 Army School of Health, Latchford Barracks, Bonegilla, VIC, Australia, 4 Department of Defence, Canberra, ACT, Australia, 5 Land Division, Defence Science & Technology Group, Edinburgh, SA, Australia, 6 School of Psychology, The University of Sydney, Sydney, NSW, Australia

☯ These authors contributed equally to this work.
‡ These authors are senior authors on this work.
* rohan.walker@newcastle.edu.au

**Data Availability Statement:** All relevant data are within the manuscript and the Supporting Information file.

## Abstract

This paper describes the conceptual design of a virtual reality-based stress management training tool and evaluation of the initial prototype in a pilot efficacy study. Performance Edge virtual-reality (VR) was co-developed with the Australian Defence Force (ADF) to address the need for practical stress management training for ADF personnel. The VR application is bio-feedback-enabled and contains key stress management techniques derived from acceptance and commitment and cognitive behavioural therapy in a modular framework. End-user-provided feedback on usability, design, and user experience was positive, and particularly complimentary of the respiratory biofeedback functionality. Training of controlled breathing delivered across 3 sessions increased participants' self-reported use of breath control in everyday life and progressively improved controlled breathing skills (objectively assessed as a reduction in breathing rate and variability). Thus the data show that a biofeedback-enabled controlled breathing protocol delivered through Performance Edge VR can produce both behaviour change and objective improvement in breathing metrics. These results confirm the validity of Performance Edge VR platform, and its Controlled Breathing module, as a novel approach to tailoring VR-based applications to train stress management skills in a workplace setting.

## Introduction

Many jobs and workplaces include unavoidable exposure to situations in which a threat (real or perceived) is present and is to some degree uncontrollable and/or unpredictable. This can

**Funding:** Funding for this project was provided by the Australian Defence Innovation Hub (https://www.innovationhub.defence.gov.au/; Contract CTD 2017-5; authors EN & FRW). The funders had no role in study design, data collection and analysis, decision to publish, or preparation of the manuscript.

**Competing interests:** All authors were involved in the development of the Performance Edge VR application. Intellectual property relating to the application is owned by The University of Newcastle. The Australian Defence Innovation Hub provided funding for application development and retains rights for its future use. This does not alter our adherence to PLoS ONE policies on sharing data and materials.

range from the extreme threat encountered by a soldier clearing a building of hostile combatants, to a sales consultant dealing with a hostile or aggressive customer. Under these circumstances, the presence of threat and low levels of controllability and predictability are core elements that can trigger a biological stress response [1, 2].

The acute stress response involves the co-ordinated engagement of a variety of biological subsystems, and provides the impacted individual with increased biological capacity to manage the threatening situation. When the stress response is engaged acutely and is appropriate to context, it can have a range of beneficial effects, including: enhanced attention, decreased response times, priming of immune responses and sharpened memory formation [3–6]. However, in some circumstances the effects of the stress response can be detrimental. This may occur when engagement of the stress response occurs at non-optimal levels (relative to the circumstances) or when stress is chronic. Under these conditions, the stress response can compromise attention, reduce situational awareness, impair judgement and decision-making, as well as provoke a variety of other negative consequences [7, 8]. These detrimental effects can further increase the severity of the stress event due to the way the primary behavioural response is handled. Importantly, training can allow individuals to modulate their responses to stressful situations.

Effective stress management skills are beneficial in both everyday life and workplace settings. Many organisations, rather than focusing on how to handle stress, instead focus on the avoidance of stress-provoking situations. Training tends to be orientated towards the iterative development of fundamental skills with exposure to increasingly realistic training scenarios as competencies develop. When stress management skills training does occur, it most commonly focusses on the development of 'external' skills. For instance, in a military context, emphasis is typically placed on physical fitness, competency in (and knowledge of), tactical procedures and equipment, workflow routines and rules of engagement. "Inner world" skills are those that allow trainees to shape their own inner states including (but not limited to): skills to manipulate learning, memory, attention, cognition and emotion. The broad process of developing inner world skills was recently defined by Aidman as 'Cognitive Fitness' [9]. Including psychological 'inner world' skills training to support preparation for arduous conditions has been explored for military trainees [10, 11].

There is robust evidence demonstrating that inner world skills can be trained and that training can modulate components of the stress response [12–14]. This is particularly well-developed for the use of breath control to modulate stress responses, particularly when supplemented with biofeedback [15]. Voluntary respiratory control is a common and well-validated technique for reducing stress and anxiety [16–18], reducing blood pressure [19, 20], and as a therapeutic intervention for psychophysiological disorders [21]. The hallmarks of controlled breathing include intentional, deep and rhythmic breathing at a reduced rate, measurable by a reduction in respiratory rate and its variability. This type of breathing has a range of physiological effects on respiratory muscle activity, ventilation efficiency, chemoreflex and baroreflex sensitivity, heart rate variability, blood flow dynamics, respiratory sinus arrhythmia, cardiorespiratory coupling and sympatho-vagal balance (reviewed in [22]). Further, it is used in martial arts training, yoga and meditation practices. Whilst the efficacy of these approaches is widely believed to be purely anecdotal, there is increasing interest in documenting effects in structured studies, including for psychological training in the military and first responder organisations [23–25]. A systematic review of 15 studies assessing slow breathing techniques demonstrated that these approaches improved heart rate variability, respiratory sinus arrhythmia and central nervous system activity [26]. These changes were linked with increased reporting of comfort, relaxation, vigour and alertness, as well as with decreased arousal, anxiety, depression, anger and confusion [26]. Links between respiratory and emotion control were

recently reported in [27], providing evidence of a potential mechanism in the brain underlying the documented benefits of controlled breathing.

To date, structured delivery of "inner world" training as a standalone skill in stress management, independent to a primary training goal such as physical fitness, yoga or martial arts, is limited. The concepts are often delivered as an introduction, which is adequate from the perspective of orientation but less useful for skill development and consolidation. More advanced practical training, particularly in a work-place setting, is commonly delivered as preventative secondary stress management in individual-oriented and facilitator-based one-on-one sessions after stress has been identified and often after a stress-linked psychological disorder has developed [28–31]. The challenge with providing intensive, expert-delivered, inner-world skill training in a prospective manner to *ab initio* trainees is the cost of service provision, which significantly limits the practicality of this model. As such, it is important to identify scalable training solutions that can be broadly implemented in workplace environments, as part of standard training processes.

Training supported by digital platforms can address the issue of scalable deployment of inner world training. These platforms range from desktop or smartphone applications, through to mixed-reality (XR) approaches, a term encapsulating virtual- and augmented reality (VR/AR) and 360-degree video. Several investigations have explored the utility of VR to alter 'inner world' states as they provide additional immersion and privacy compared to 2D screen applications. Of note, the preponderace of research focus on using a virtual environment to 'inoculate' trainees with stressful experiences, rather than training and developing a specific skill set to manage the stress (e.g. VR-based stress exposure treatment for post-traumatic stress disorders; reviewed in [32]). Specifically, in the context of military training, 14 studies using VR-based applications focussed on exposure to stressful situations or passive relaxation techniques (e.g. meditation), rather than specific skills training [33]. Several VR applications have integrated biofeedback including heart and breathing rate into a virtual environment [34–38], with reported improvements in user engagement and motivation. To date, these applications have not integrated respiratory biofeedback into a structured skills training program for a military audience.

In collaboration with the Australian Defence Force, our research Centre is co-developing a set of empirically-validated, VR-based training tools for stress management. This platform, referred to as Performance Edge VR, is a novel, modularised training package for stress management skills. Each module addresses a different stress management approach or technique. The current manuscript provides an overview of the conception, design and initial testing of the Performance Edge VR platform (user interface and user experience), with a specific evaluation of the first module, which focusses on controlled breathing training as a stress management skill supported by integrated respiratory biofeedback. The framework for the platform was guided by principles established in the Australian Defence Force's existing BattleSMART stress management and resilience training package [39]. An initial pilot study was conducted to evaluate the overall ability of the developed prototype to address training outcomes and gather critical information on general acceptability and usability of the technology, subject matter and design choices.

## Methods

### Project conception and scoping

The Performance Edge VR application resulted from initial discussions between the Centre for Advanced Training Systems (ATS; https://www.advancedtrainingsystems.org.au) and Army Psychologists within the Department of Defence, with input and advice from the Defence

Science Technology Group. Initial meetings identified the potential utility of a VR-based application providing stress management skills training for deployment within the Australian Defence Force. Priority was placed on an application to support the development and consolidation of 'basic skills' for trainees.

## Content development, review and approvals

The project was developed using an Army-initiated co-design approach. Specifically, the content, structure, tone, philosophy and outlook of the tool were defined by input provided by the Australian Army, Joint Health Command and the Defence Science Technology Group. ATS facilitated practical creation, implementation and research evaluation of the tool. Two external clinical psychologists were involved in the content development process. The subject matter was informed by Acceptance & Commitment Therapy (ACT) principles, which are based on extensive research evidence (meta-analysis by [40]). Critical members of the Defence organisation were consulted to review and amend content with final authority provided by the Director, Mental Health Strategy and Research within Joint Health Command.

## Virtual reality application development

Application development proceeded via an iterative, step-wise approach including 4 phases: 1) scoping and concept creation; 2) content refinement and development; 3) beta-version testing with user feedback; and 4) finalisation of the prototype. A software developer (Jumpgate VR, Adelaide, Australia) was contracted to code the Performance Edge VR application for the Oculus Rift headset on a Unity engine platform.

## Virtual reality & biosensor hardware and software

Performance Edge VR was developed for the Oculus Rift VR platform, which requires a direct cable connection to a high-performance gaming laptop (minimum specifications: Intel i5-4590 or greater, NVIDIA GTX 1060 video card or greater, 8 GB+ RAM, Windows 10 and compatible video output; used in current study: Alienware Dell 15R3, Intel core i7-7700HQ, CPU@2.8 GHz, NVIDA GTX 1080, 16 GB RAM, Win 10 Pro) and external VR positioning sensors. Respiratory rate measurements and in-application biofeedback were provided by an EquiVital biosensor respiratory harness (Hidalgo, UK) with integrated transmitter (SEM), with the digital signal transmitted wirelessly via Bluetooth to the laptop. Intellectual property relating to Performance Edge VR is owned by The University of Newcastle. If interested in accessing Performance Edge VR application for research purposes, please contact the corresponding author.

## Pilot efficacy study

A pilot efficacy study was undertaken to address two aims. Firstly, to collect initial user feedback on user experience, user interface, graphics, design and the suitability of hardware and software for the overall Performance Edge VR application, as well as for aspects of the module for controlled breathing specifically. Second, we aimed to assess the effects of the Performance Edge VR controlled breathing training module on breathing rate and variability and user-reported breathing practices. The study protocol was approved by the Human Research Ethics Committee of The University of Newcastle (H-2019-0037). Informed consent was provided by all participants at the beginning of their first visit.

Participants were invited to attend 3 training sessions using the Performance Edge VR Controlled Breathing module. Each session was performed on a single day, over approximately 90 minutes, and consisted of pre-training breathing evaluation (10 minutes), completion of the

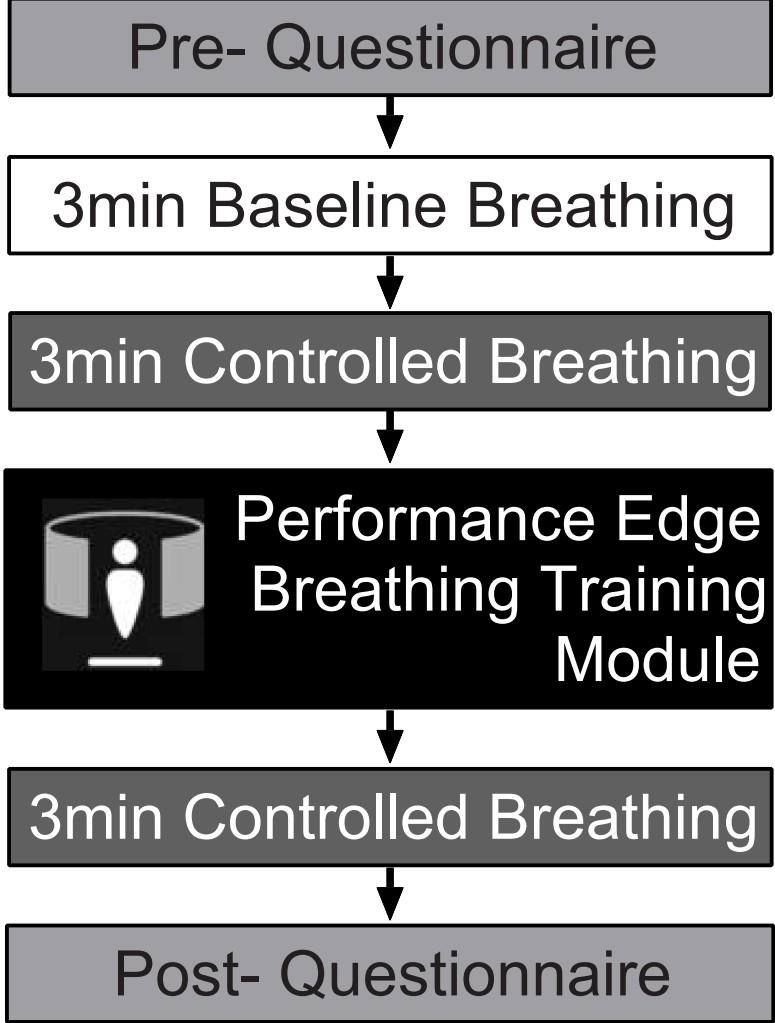

**Fig 1. Pilot efficacy trial session design.** Participants attended 3 training sessions, each held approximately 1 week apart. Each session consisted of pre-training questionnaires, followed by baseline breathing data collection (baseline and controlled). Participants then completed the Performance Edge VR Controlled Breathing module. After training, controlled breathing was again assessed and participants completed post-training questionnaires. All breathing data was captured using the Equivital system.

Performance Edge VR training module (25–30 minutes) and post-training breathing evaluation and questionnaires (25 minutes). Each of the 3 sessions were conducted approximately 1 week apart. Fig 1 provides a schematic overview of the components included in each session.

### Study participants & recruitment

Although the intended audience for the Performance Edge VR program is *ab initio* military trainees, a convenience sample of participants from The University of Newcastle were

**Table 1. Pilot trial participant characteristics (n = 30).**

| | |
|---|---|
| **Total participants** (Male / Female) | 30 (13 /17) |
| **Age** (Mean ± SD; Median) | 30.5±7.9; 30 |
| **Previous Experience with Controlled Breathing** | 50% |
| **Nijmegen Score** (Mean ± SD) | 5.2±4.8 |

Baseline questions relating to previous experience with controlled breathing included yoga and meditation as examples.

recruited to assess overall effectiveness, acceptability and usability. A total of 30 healthy, young volunteer participants were recruited for the pilot study between August 2019 and February 2020, via broad email distributions to University email lists and flyers posted at the Callaghan campus (The University of Newcastle). Participants with self-reported physical conditions (e.g. heart condition, asthma, mobility-impairment), psychological conditions (e.g. anxiety, depression, PTSD), epilepsy or potential breathing dysfunction (identified by the Nijmegen questionnaire, as described below) were excluded from the study. Intention was to recruit an unbiased participant group representative of the general population, rather than military trainees *per se*, while excluding factors that may impact on objective breathing assessments. Participant demographics are presented in Table 1.

## Study protocol

At the beginning of the first session, participants were invited to provide informed consent and complete two pre-study questionnaires. One questionnaire captured relevant participant demographics (including gender, age and previous experience with controlled breathing exercises). Participants also completed the Nijmegen questionnaire, a validated assessment tool to identify functional respiratory dysfunction [41]. The Nijmegen questionnaire was used for screening to exclude any participants with potential underlying respiratory conditions, which could potentially interfere with assessments of breathing rate/variability and completion of the training exercises. A Nijmegen score >19 was set as a cut-off for participant exclusion, and no participants exceeded this score.

At each session, baseline breathing patterns were objectively recorded using the Equivital Life Monitor Belt (EquiVital, Hidalgo, United Kingdom) and LabChart 8.0 software (ADInstruments, Australia), prior to commencement of VR training. The assessment included two tasks: 1) 3 minutes of baseline breathing, where participants read a manuscript without vocalizing to minimise distractions, and 2) 3 minutes of "controlled breathing", where participants were instructed to concentrate on their breathing and attempt to reduce it to their lowest comfortable rate. Real-time digital filtering using a triangular (Bartlett) smoothing window with a window width of 205 samples was applied to minimise noise and movement artefacts. Respiratory rate (cycles per minute) and variability (coefficient of variation) were calculated using the LabChart internal analysis tool. Due to the effects of low-pass filtering and smoothing, respiratory depth/amplitude was not assessed in this analysis.

After completing baseline breathing measurements, participants were fitted with the Oculus Rift VR headset (Oculus VR, USA). They were then instructed to complete the Performance Edge VR Controlled Breathing module, following on-screen prompts.

After completing VR training, controlled breathing measurements were collected for 3 minutes, when participants were again asked to maintain their lowest comfortable breathing rate. Participants also completed post-training questionnaires, which sought feedback on VR

hardware and software (session 1) and skills acquisition (session 3). Question lists are provided in S1 File.

## Statistical analysis

Statistical analysis was performed using Prism v.8 (GraphPad, USA). Two-way ANOVA for repeated measures adjusted for multiple comparisons was used to compare pre- and post-respiratory measures across multiple sessions. One-way ANOVA for repeated measures adjusted for multiple comparisons was used to compare pre-respiratory measures across multiple sessions. Changes in respiratory rate and variability were analysed using paired t-tests. $P < 0.05$ were considered statistically significant for all analyses.

## Results

### Development and concept design of the Performance Edge VR training platform

Initial conception of the entire Performance Edge VR platform started with an extensive internal development process, involving members of the ATS team, clinical and military psychologists and input from key military stakeholders based on existing evidence. The suitability of a modular framework was identified (screenshots; Fig 2A), to support focus on individual stress management skills in sequence with flexibility for training delivery while acknowledging that users start with different levels of existing knowledge / skills. Each module was designed to contain 3 core components: 1) theoretical knowledge, 2) skill acquisition and 3) skill

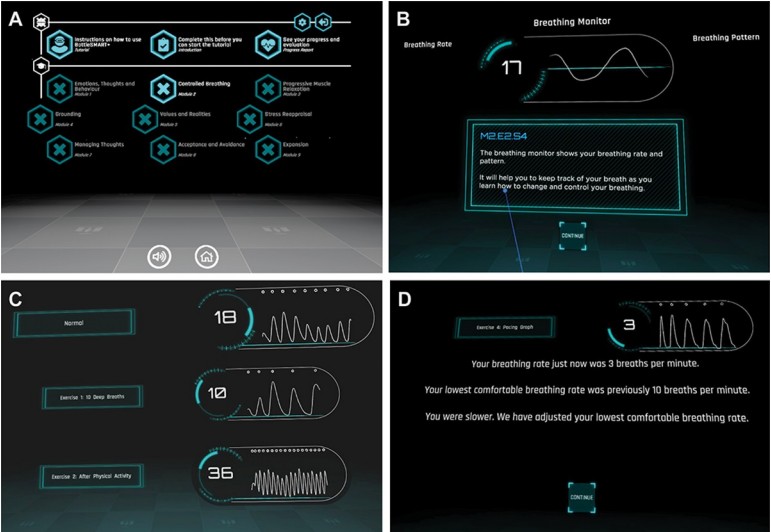

**Fig 2. Screenshots and key features of Performance Edge VR.** A) Screenshot of the main menu design, which provides access to 3 overview items (top) and 9 training modules (bottom). The main menu can be accessed at any time and includes a general introduction to Performance Edge VR, a tutorial on how to navigate and use the application and the Progress Reporting environment. B) Screenshot of the respiratory trace breathing monitor, which includes a breathing trace and average breathing rate. Live biofeedback is a key feature of the Controlled Breathing module, which allows users to visualise their breathing pattern and rate, in order to establish their lowest comfortable breathing rate. C) Initially biofeedback is provided to users after completing exercises, to promote awareness of breathing pattern and visualise how breathing changes under different conditions (unprompted baseline "Normal"; after 10 deep breaths; after mild exercise [standing & sitting 10 times]). D) In later exercises, a live respiratory trace and rate are used to visualise and track breathing to support the establishment of their lowest comfortable breathing rate. © Screenshots captured from Performance Edge VR, developed by The Centre for Advanced Training Systems.

consolidation. At completion, Performance Edge VR will include 8 discrete modules, including: 1) Emotions, Thoughts and Actions; 2) Controlled Breathing; 3) Progressive Muscle Relaxation; 4) Grounding; 5) Values and Realities; 6) Stress Reappraisal; 7) Managing Thoughts / Cognitive Defusion; and 8) Acceptance and Avoidance. Module 2 (Controlled Breathing) was identified for initial development and is the primary focus of the current manuscript.

Software development occurred in stages, consisting of: 1) scoping; 2) content refinement; 3) beta-version testing with feedback; and 4) prototype finalisation. Specific design decisions were informed by identified training goals and iterative rounds of user feedback.

The primary focus of the application is delivery of practical evidence-based stress management skills training, and we were conscious of making the content as engaging and interactive as possible. As such, elements of gamification were integrated to provide a sense of personal achievement, performance improvement and enhance user engagement (Fig 2D). User feedback and performance tracking is incorporated into the platform and applied across all modules. A freestanding and accessible user reporting environment allows users to track their progress and scores / performance across individual exercises and modules.

Considerable time was spent on the design of graphics and user-experience. A futuristic design was applied, which paid homage to space-themed computer games (Fig 2). This design aligned with the relative novelty of VR technology and was proposed to promote engagement and interest from the target user audience (i.e. a predominantly gaming-literate, largely male populations in their early 20s). It was noted that while the initial target audience was military personnel, visual design features were created using a neutral palette without specific reference to the military context. This was in part due to the military tri-service specific branding but also allowed emphasis that the skills covered in Performance Edge VR are relevant for any stress-provoking situation.

## Internal user testing & feedback

Module beta-versions underwent multi-stage internal and external testing following an iterative development of the final version. Feedback was sought from all Army and psychologist contributors and Australian Defence Force stakeholders. While Performance Edge VR was conceived as a tool based on ACT psychological interventions and pre-existing BattleSMART content, the final prototype does not look or "feel" like either a mental health tool or military training system *per se*. Evidence-based and well-established exercises were re-branded to be delivered in a virtual environment with a highly task-oriented training flow and distinct language, graphics and visuals. It became apparent during this stage that even short moments of inactivity in the virtual environment result in users disconnecting from the experience. Thus, substantial redesign occurred to reduce verbal and written instructions to a minimum. The application was designed to take advantage of the intuitive nature of the VR environment and to place users directly into actionable exercises with minimal instructions or additional information. Feedback also informed the development of an exercise format whereby each exercise progressively builds on the preceding exercise, becoming more challenging over the course of each module.

User feedback as well as feedback from the study coordinator highlighted several potential issues with the existing hardware as potential barriers for broader implementation. While the EquiVital respiratory belt captured accurate respiratory data and was suitable for use in a research setting, the Bluetooth connections between the SEM and computer was variable, with impacts on maintaining biofeedback throughout training. Further, the Oculus Rift headset is tethered by a cable to the laptop, which limits user mobility. While these issues could be

overcome in a 1-on-1 research setting, the set-up and connection of multiple systems in close proximity during implementation is a practical barrier that must be overcome for broader future dissemination.

## A Performance Edge VR training module for controlled breathing skills

The Controlled Breathing module contains 7 exercises of escalating difficulty. Within the module, users are oriented to a visual display of their respiratory trace, integrating biofeedback data as a respiratory monitoring tool (Fig 2B and 2C). In the first instance, the respiratory monitor is used to visualise changes in their personal respiratory rate and trace in different situations; stationary, after minimal exertion (standing up / sitting 10 times) and when focussing attention on their breathing (Fig 2C). Exercises then focus on the development and maintenance of a consistently slow and steady respiratory rhythm, termed the 'lowest comfortable breathing rate'. While completing exercises, trainees gain awareness and practice control over their breath by first using the live respiratory trace as a visual guide. Escalating training complexity is achieved by removing the visual aid, introducing a distracting environment (a 360-degree rock concert video) and by engaging in a physical task (a crossbow shooting exercise, where target speed is dictated by breathing rate). Individual performance on each exercise (specifically the ability to maintain their previously established lowest comfortable breathing rate) is captured and provided to the users via direct feedback after each exercise (Fig 2D). Performance data is also stored and viewable via a reporting environment accessible from the main menu.

## Pilot study recruitment and participant information

A structured pilot study of the completed Module 2 was undertaken, to seek user feedback on the Performance Edge VR application overall and the controlled breathing module specifically. This study was also designed to determine whether training improved user self-reported and objective measures of controlled breathing performance. A total of 30 participants were recruited to perform training in 3 separate sessions over approximately 3 weeks. Participant demographics are presented in Table 1. Three potential participants opted not to participate after reading the participant information statement, due to the length of study requiring attendance of 3 separate study visits. However, all participants that consented to commence the study completed all 3 study visits. All available participant data was included in analysis (although we note that subjective data from a single visit by one participant was missing and not included in analysis). Participants had a mean age of 30.5±7.9 years (mean±SD; median = 30). Half of all participants reported previous experience with controlled breathing at the beginning of session 1 (e.g. yoga, meditation) and very low levels of self-reported respiratory dysfunction (mean Nijmegen score = 5.2±4.8; >19 = cut-off score for participant exclusion).

## Participant feedback on Performance Edge VR hardware and software

To gain feedback on the hardware used, design choices, modular structure and usability of Performance Edge VR overall, participants were asked to complete post-training questionnaires rating the application. Participants were very positive about the comfort of individual hardware components, as well as overall, with weighted averages for all categories >4 (1 = extremely uncomfortable, 5 = extremely comfortable; specific averages were VR headset = 4.1, biometric belt = 4.4, headphones = 4.3, controllers = 4.7, overall comfort = 4.3; Fig 3A). Similarly, participants provided unanimously positive feedback on aspects of user experience and software, including appearance / design, instructions, intuitiveness, navigation and

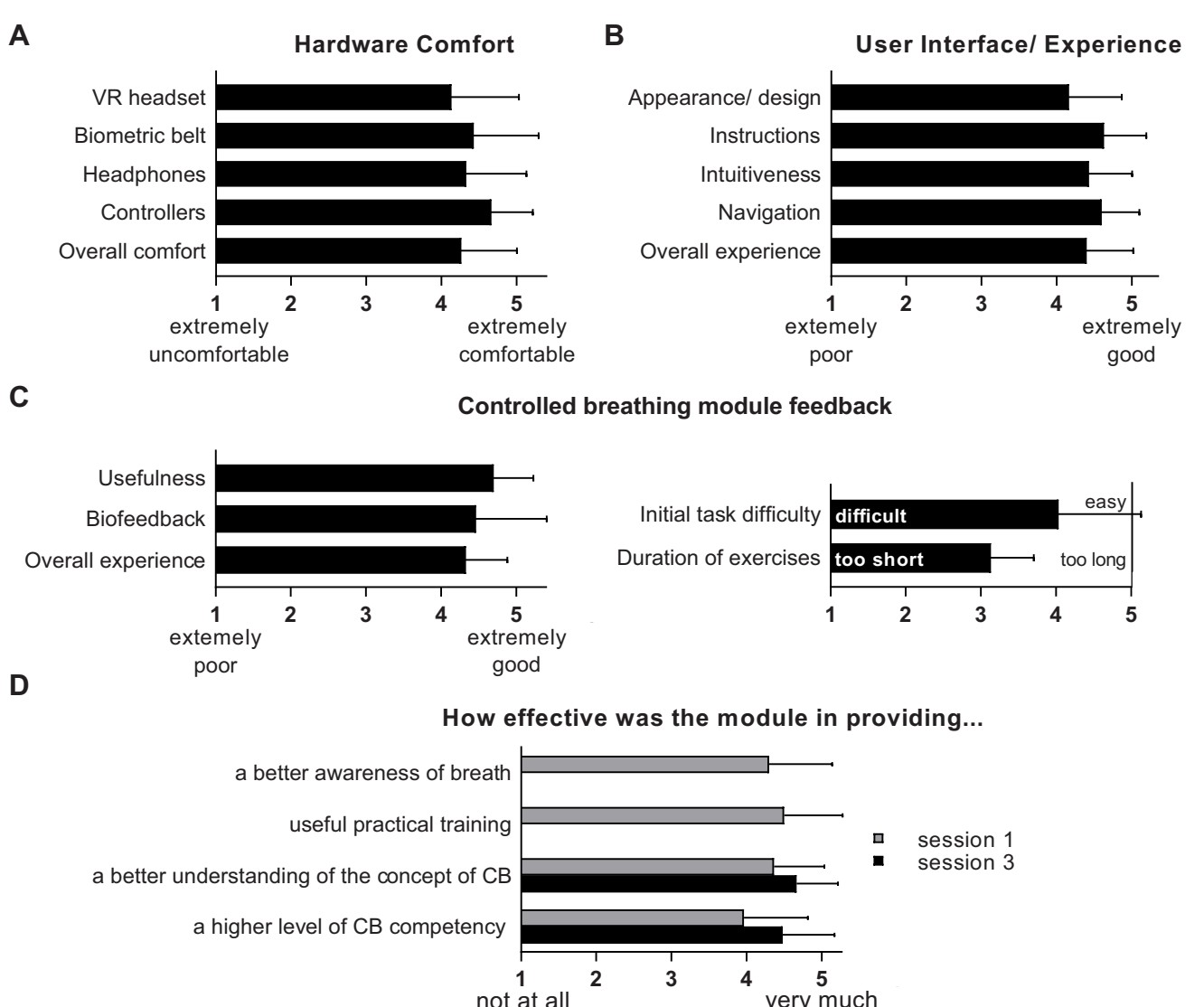

**Fig 3. User survey feedback on Performance Edge.** Participants provided feedback before and after using Module 2 –Controlled Breathing, across 3 training sessions. Survey responses were collected for the entire Performance Edge VR platform after session 1, relating to A) hardware comfort and B) software components and user interface / experience. Specific to Module 2, users provided feedback on C) perceived utility, biofeedback and overall experience, as well as task difficulty and appropriateness of durations. D) Users also provided feedback on perceived effectiveness of training on aspects of breathing awareness, usefulness, understanding and competency after session 1 (and after session 3, where indicated). Data presented as weighted averages based on a 5-point Likert scale (n = 30); mean±SD.

overall experience (weighted averages = 4.4, 4.2, 4.6, 4.6, 4.4, respectively; 1 = extremely poor, 5 = extremely good; Fig 3B). Two participants reported neck strain due to headset weight. No other user safety and/or tolerability issues were identified, and all participants were able to complete all VR training sessions.

## User feedback on controlled breathing module features and training exercises

Participants also provided specific feedback on the Controlled Breathing content and delivery. Participants found the module useful (weighted average = 4.7) and rated the overall experience as good (weighted average 4.3; 1 = extremely poor, 5 = extremely good; Fig 3C). Further, the

inclusion of biofeedback was perceived as very helpful (weighted average 4.5; 1 = not at all helpful, 5 = extremely helpful; Fig 3C). Regarding individual training exercises, duration was considered satisfactory (weighted average = 3.1; 1 = too short, 5 = too long) and difficulty was on the 'easy' end of the spectrum (weighted average of 4.0; 1 = difficult, 5 = easy).

We further assessed the perceived effectiveness of training, across a range of outcomes (Fig 3D). After the first session, participants agreed that the module effectively improved general awareness of their breath (weighted average = 4.3; 1 = not at all, 5 = very much), provided a useful practical training opportunity (weighted average = 4.5), improved understanding of the concept of controlled breathing (weighted average = 4.4) and increased perceived controlled breathing competency (weighted average = 4.0; Fig 3D). Perceived effectiveness of understanding controlled breathing and competency were further increased following the 3rd training session (weighted averages = 4.7 & 4.5, respectively; Fig 3D). Thus, in addition to providing positive feedback on the specific details of the breathing module delivery and design, participants also reported effective knowledge transfer and increased perceived competency in controlled breathing.

## Self-reported behaviour change effects of controlled breathing training under Performance Edge VR

Participants were queried on whether they would (or were) applying controlled breathing techniques beyond the study context, to assess self-reported behaviour change. Participants agreed they were likely to use controlled breathing when encountering a stressful event after completing the 1st training session, which increased to "very likely" following the 3rd training session (weighted average session 1 = 4.1. session 3 = 4.6; 1 = probably not, 5 = very likely; Fig 4A). Participants also reported increased attention to their breath after completing multiple training sessions (Fig 4B) and increasing use of controlled breathing skills between training sessions, which increased over multiple training sessions (Fig 4C). General awareness of breath was increased in 77% of participants in the week following the 1st training session and in 97% of participants in the week after the 2nd session (Fig 4B). In contrast, no participant reported a reduction in their attention to breath after training (Fig 4B). Similarly, 83% of participants

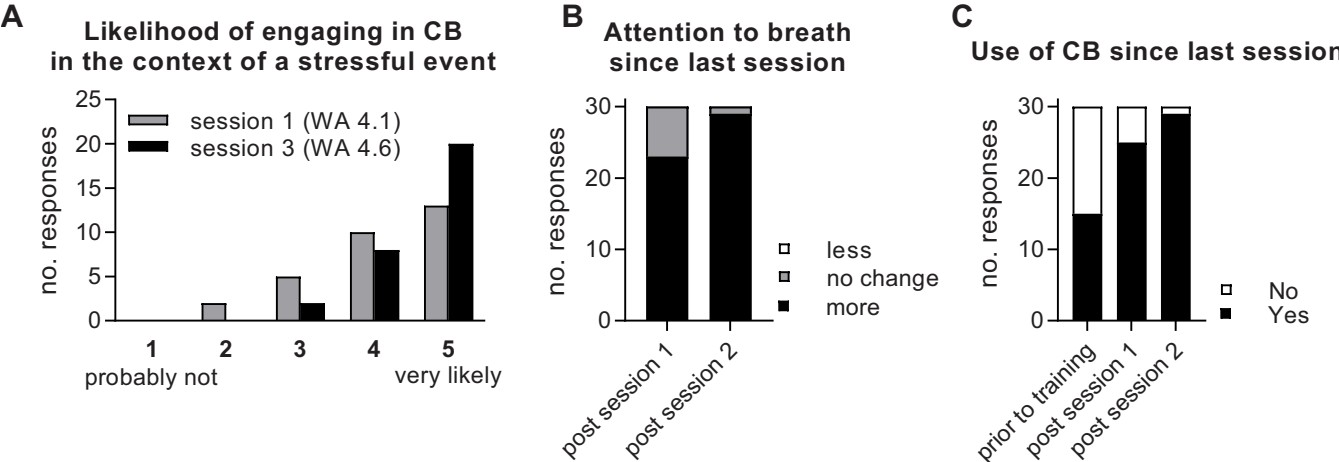

**Fig 4. User-reported effects of Performance Edge VR training on controlled breathing behaviours.** Users self-reported A) likelihood of engaging in controlled breathing in response to a stressful event in the future and B) their level of attention to their breathing in the week between sessions. C) Participant-reported use of controlled breathing before initiating training and between training sessions. Data presented on a 5-point Likert scale (WA = weighted averages) and individual responses (n = 30).

used controlled breathing in the week following session 1 and 97% in the week following session 2 (Fig 4C). Thus, participants self-reported behaviour change following VR training, with increased awareness of breath and increased application of controlled breathing skills.

## Performance Edge VR training reduced objective measures of controlled breathing including both respiratory rate and variability

We also quantified the effects of VR training on objective measures of breathing, including respiratory rate and variability. Breathing traces were collected during baseline breathing and for pre- and post-training controlled breathing (representative traces Fig 5). When prompted to engage in "controlled breathing", respiratory rates were significantly reduced after each training session, compared to pre-training rates recorded at the same session (Session 1 MD = -2.3±0.5 bpm; Session 2 MD = -1.1±0.3 bpm; Session 3 MD = -0.5±0.2bpm; MD = mean

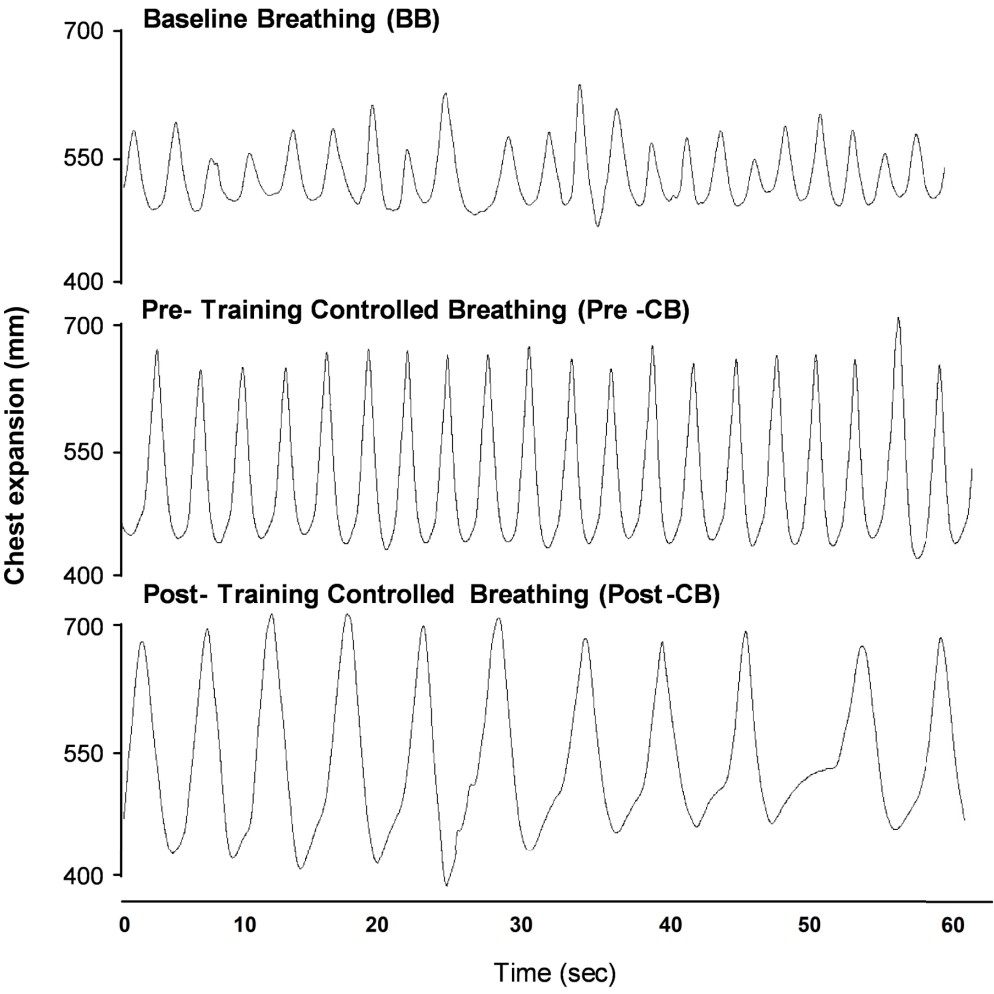

**Fig 5. Representative biometric respiratory recording.** Representative data captured using LabChart software and an Equivital respiratory harness, recorded at baseline (BB) and after being prompted to engage in controlled breathing task before (Pre-CB) and after training (Post-CB).

**Training effect per sessions**

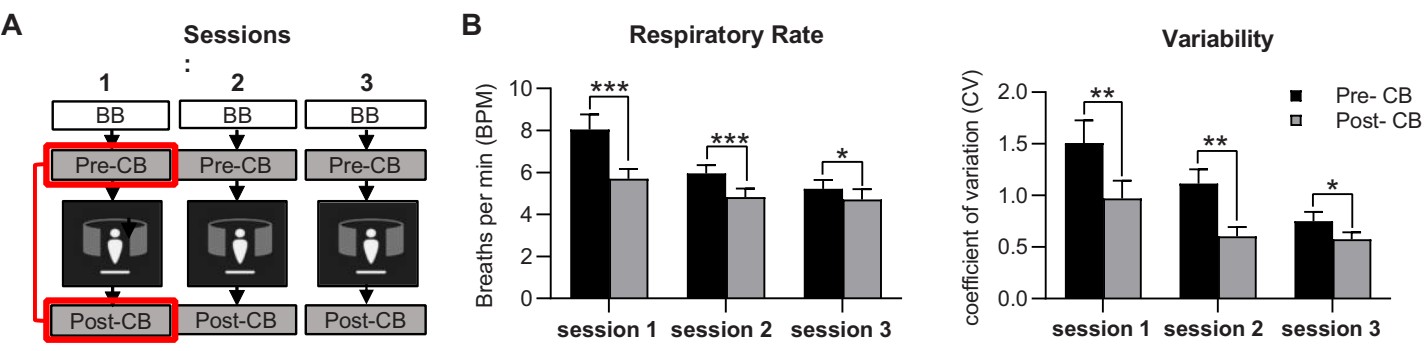

**Training effects between sessions**

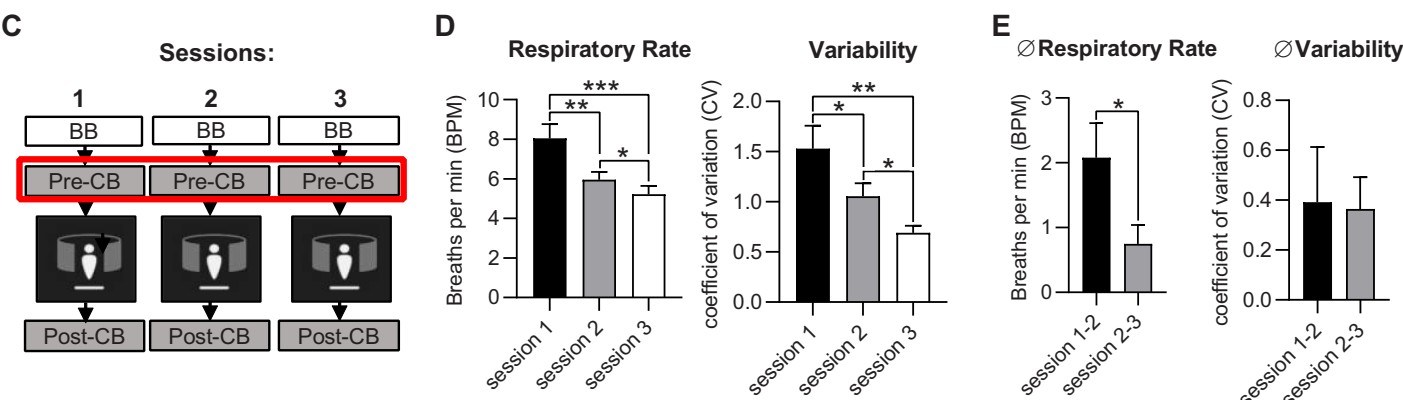

**Fig 6. Objective respiratory measures improve with VR-based training.** Average respiratory rate (breaths per minute) and respiratory rate variability (standard deviation of rate) collected before (pre-CB) and after training (post-CB) for each session, presented to support comparisons for A/B) each session and C/D) between sessions. E) Differences in pre-training respiratory rate and variability between sessions. (n = 30; data presented as mean±SEM; *p<0.05, **p<0.01, ***p<0.001) BB = Baseline Breathing, CB = Controlled Breathing.

difference ± SEM; Fig 6B). Similarly, respiratory rate variability was significantly reduced across each training session (Session 1 MD = 0.5±0.2 bpm; Session 2 MD = 0.3±0.1 bpm; Session 3 MD = 0.2±0.1 bpm; Fig 6B).

In addition to decreased respiratory rates and variability across each session, respiratory rates progressively decreased across multiple training sessions, as assessed during pre-training (Session 1–2 MD = 2.0±0.5bpm; Session 2–3 MD = 0.7±0.3bpm; Fig 6D). Similarly, respiratory rate variability was significantly reduced across multiple sessions (Session 1–2 MD = 0.5±0.2; Session 2–3 MD = 0.4±0.1; Fig 6D). Differences in respiratory rate between session 1 and 2 exceeded those observed between sessions 2 and 3 (Fig 6E), but decreases in respiratory rate and variability were observed between all sessions assessed. Thus, VR training reduced respiratory rate and variability during controlled breathing, assessed using objective measures. This training effect was retained between training sessions, with additional improvement observed across 3 training sessions.

As 50% of participants indicated previous experience in controlled breathing prior to the first Performance Edge VR training session, we performed a post-hoc analysis grouping participants by experience. Although this discovery-focused analysis was underpowered, the data

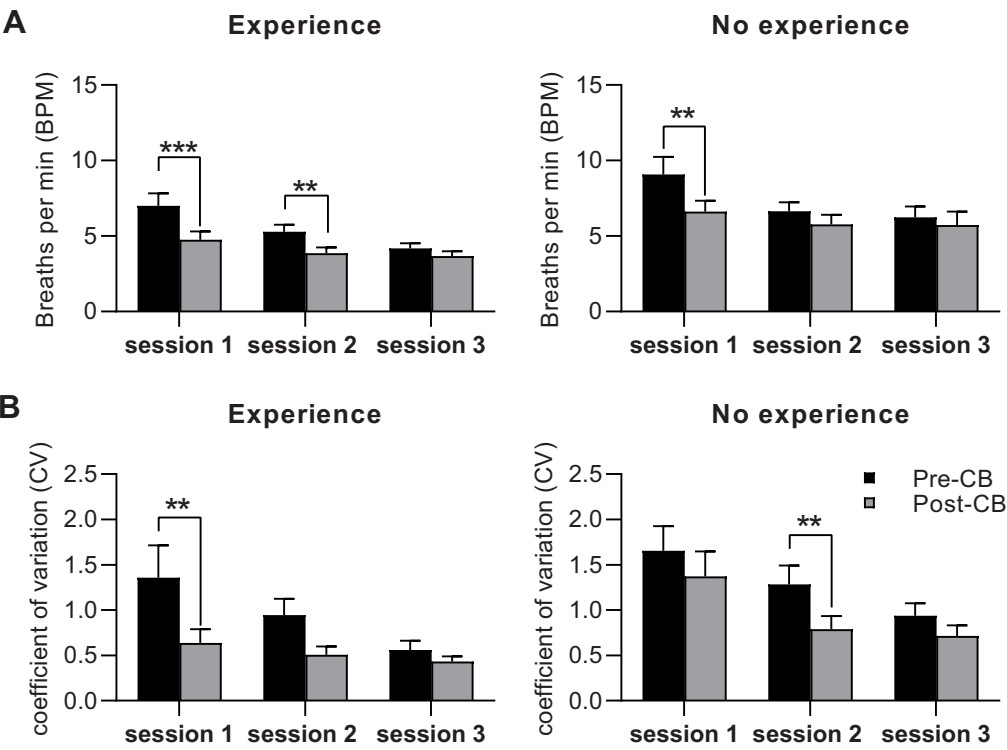

**Fig 7. Objective respiratory measures improve irrespective of previous experience in controlled breathing.** A) Average respiratory rate (breaths per minute) and B) respiratory rate variability (standard deviation of rate) collected before (pre-CB) and after training (post-CB) for each session, grouped based on participant-reported previous experience with controlled breathing. Post-hoc analysis (n = 15 per group; data presented as mean±SEM; *p<0.05, **p<0.01, ***p<0.001) CB = Controlled Breathing.

indicates a reduction in respiratory rate and variability between pre- and post-training for both groups (Fig 7). Both groups of participants with and without experience in controlled breathing reduced breathing rates after training (Session 1: experience MD = -2.2±0.7 bpm; no experience MD = -2.6±0.9 bpm; Fig 7A).

## Discussion

This report describes the initiation and concept design of a biofeedback-enabled virtual reality program to deliver practical stress management skills training, termed Performance Edge VR. The project is co-developed between the University of Newcastle, the Centre for Advanced Training Systems (ATS) and the Australian Army Office of Strategic and Operational Mental Health, to address a gap in delivery of repeated, practical training of stress management skills within the Australian Defence Force. In this study we evaluate the initial prototype usability, its training efficacy of controlled breathing as a stress management skill and overall user feedback and acceptance of the technology, subject matter and delivery style. Objective data provided preliminary evidence that the application facilitates effective controlled breathing skill development after a single training session, which increased with 3 consecutive training sessions and was retained for at least one-week post-training. Survey data supported positive training outcomes, indicated desirable behavioural change and successful knowledge transfer. General feedback was positive for all application components including hardware and software, design features and usability. The Performance Edge VR prototype and virtual reality

technology were perceived as a suitable and effective modality to facilitate practical stress management training, paving the way to further advance the development of the program.

## Performance Edge prototype as a tool to train controlled breathing skills

This first Performance Edge prototype contains all core structural elements of the modular designed training application and one of eight functional training modules. The controlled breathing module is a core component of Performance Edge VR and aims to deliver both background knowledge on the benefits and appropriate usage of controlled breathing for stress management, as well as skill development and practical training. Both quantitative and qualitative data indicate that the Performance Edge VR prototype facilitates knowledge transfer and effective training of controlled breathing. Effectiveness of training includes improvement in controlled breathing skills, defined as a reduction in respiratory rate and variability. Improvements were identified after a single training session, with progressive improvement over 3 consecutive training sessions (one week apart), skill retention after one week, and a gain in self-reported skill competency as well as behavioural change. Our post-hoc data analysis, despite not being powered sufficiently, strongly suggest that the training benefit on breathing metrics occurs for both experienced and in-experienced trainees. Of note, in this study we did not clarify or capture the level of competency for trainees who reported previous experience in controlled breathing. Future studies are therefore required to investigate the training benefit for users with different levels of existing skill competency.

Intentional modification of breathing/controlled breathing is a common approach for managing stress and a tool used in Cognitive Behavioural Therapy (CBT). Different types of breathing techniques and styles, including attentional breathing, diaphragmatic breathing at a consistent breathing rate and breath focused meditation have all been investigated as effective preventative interventions on the stress response [42–46]. Skills training of breath control has also been evaluated in relation to workplace stress, particularly within the police force [44, 47, 48].

Lehrer & Vashillo propose breathing at resonance frequency around 0.1 Hz (4.5 to 6.5 breaths/min), which facilities respiratory sinus arrhythmia (a beneficial physiological phenomenon) via resonance between respiratory and cardiovascular oscillations [49, 50]. Existing non-VR HRV biofeedback training tools like the StressEraser™, Helicor, USA and HeartMath, LLC emWave Pro software incorporate resonance frequency and guide trainees to find their optimal slow respiration rate [51, 52]. Studies including those within the police force, demonstrate improved performance outcomes, psychological stress measures and attention control following skills training [48, 53, 54]. Whilst certainly effective, it remains unclear whether breathing at resonance frequency has any advantages compared to other breathing techniques used in this context. A comparison of breath-focused meditation with biofeedback training targeting resonance frequency showed no difference across groups [55]. A common feature amongst all effective breathing approaches appears to be intentional and rhythmical breathing at a reduced breathing rate. In alignment with this, Performance Edge VR does not impose a specific target value but rather provides training to maintain a slow and steady breath at the lowest comfortable rate, a skill likely to positively impact the stress response and management skills.

The Performance Edge VR application features respiratory biofeedback functionality which is specifically used to visualise respiratory pattern and rate within the VR headset in real-time. As mentioned above, non-VR HRV biofeedback applications have shown to be effective training tools for attention control and general cognitive performance in a range of settings [56]. Incorporating a virtual environment with HRV biofeedback training further improved user motivation and focus [34]. Supporting a positive effect of combining biofeedback and VR on

user engagement and immersion is its increased usage in game design. For example, in the bio-feedback-integrated VR applications StressJam and Breath VR users must modify their heart-rate and breathing actions, respectively to interact with element of the gaming environment and direct gameplay [35, 36]. Like Performance Edge VR, the relaxation-focused VR game DEEP incorporates respiratory biofeedback to support diaphragmatic breathing at a slow and steady rate [37, 38]. Van Rooij et al demonstrate a self-reported reduction in anxiety state in children (aged 8–12) after playing DEEP for only 7 minutes [38]. In line with these findings, Performance Edge VR also utilises respiratory biofeedback to support gameplay and design (e.g. in the practical exercise, the movement of targets is directly linked to the user's respiratory rate). However, the primary use of biofeedback in Performance Edge VR is to facilitate structured and serious skills training of breath control via the visualisation of the live breathing pattern and rate. The breathing monitor (biofeedback component) was rated as particularly useful by users (Fig 3C). Based on this feedback, we believe the biofeedback component improved engagement and learning and was a major contributing factor to the overwhelmingly positive feedback and training success. Being able to see their breathing pattern was mentioned by almost all participants when prompted to name a component of the module they particularly enjoyed. We therefore propose that the biofeedback component was extremely useful in developing a deeper understanding of personalised breath, the changes in breath and skill development of a consistent breathing pattern and reduced rate (Fig 3).

## Delivering stress management training using virtual reality

Performance Edge VR was designed to deliver practical training of stress management skills using VR as a delivery modality. Although, VR has been used in the context of stress exposure and stress management, Performance Edge VR is unique in its attempt to provide practical training of stress management techniques. To our knowledge, Performance Edge VR is the first virtual reality-based stress management tool providing structured skills development and consolidation of commonly used preventative stress management skills. All techniques featured in Performance Edge VR are evidence-based, pooled from the existing BattleSMART training platform, combining components from CBT and Acceptance and Commitment Therapy (ACT) principles and many are commonly used in secondary prevention of stress in the workplace (meta-analysis by [31, 40]).

Existing VR applications, particularly in the military setting primarily emphasise relaxation (e.g. immersion in relaxing environments), the observation of physical responses or use repeated exposure to a stressful situation to reduce stress responses [10, 57, 58]. For example, BRAVEMIND was developed as a VR-based stress inoculation platform for the treatment of post-traumatic stress disorder (PTSD) in military personnel [59]. The subsequent Stress Resilience in Virtual Environments (STRIVE) application provides stress resiliency training in military personnel pre-deployment, through exposure to simulated combat scenarios [60]. However, unlike Performance Edge VR, the scenarios do not provide trainees with instructed, escalating exercises facilitating specific skills development to deploy during the stress scenario. The results do however indicate the ability of 'stress inoculation' to improve self-reported emotional control, particularly for the STRIVE application, and these findings motivated the inclusion of stress exposure scenarios within Performance Edge VR for skill consolidation exercises [61].

Compared to 2D screen-based delivery, virtual reality has the capacity to enhance training outcomes by increasing user immersion and focus, engagement, and privacy [62–64]. Virtual reality was selected as a suitable training modality for Performance Edge VR to address several training requirements in addition to the creation of a highly immersive training experience.

Firstly, the technology provides, cost-effective, scalable, consistent and freestanding delivery of content without the need for IT integration. Together with biofeedback, an emphasis on interactive components and personalised feedback, we hypothesise that the technology generates a sense of privacy, mimicking an individualised, facilitator-based training experience, similar to the one-on one sessions commonly used in preventative secondary stress management [31]. We further anticipated that the novelty of the technology will generate immediate interest, engagement and momentum, whilst reducing preconceptions towards the subject matter, specifically associations to yoga and/or mental health applications. We show that users were comfortable with the use of both the hardware and software and felt that the technology was suited for the delivery of practical skills training in general and stress management skills training in particular (Fig 3). Further, users enjoyed the overall training experience, particularly the interactive and immersive components, biofeedback and personalised performance feedback. This data supports the use of virtual reality as a training modality and suggest a positive contribution of technology specific features on training outcomes. Of note, no participant had to discontinue training or indicated any health complications resulting from training (e.g. nausea or motion sickness). This is relevant given that cyber-sickness is considered a major barrier to the expanded use of VR, specifically relevant for newly developed applications such as Performance Edge VR.

## Transitioning from a prototype to an effective training application

Successful creation, integration and implementation of any new training program undergoes several developmental stages and version iterations. Here we demonstrate, as a first step, the in-principle functionality, acceptance, and efficacy of the Performance Edge VR core framework: providing stress management skills training via a virtual reality interface. As our pilot study was primarily designed to evaluate the initial prototype to inform ongoing development, data was collected in a controlled laboratory setting, with a small population size (n = 30), which was recruited from a convenience sample at The University of Newcastle. Further, the intervention timeline was relatively short (3 sessions over 3 weeks) and behavioural changes were only assessed by self-report. While our findings are encouraging and support the extension of the Performance Edge VR prototype to a functional training application, the above limitations should be considered when interpreting and extrapolating the findings.

The pilot study identified the need for an updated hardware system for future versions of Performance Edge VR. Whilst the Oculus Rift headset and the EquiVital respiratory monitor delivered and addressed the intended need in a one-on-one research setting, the practical delivery of this setup within its intended training setting is unlikely to be user-friendly, cost-effective or scalable. The system has significant costs (between AU$2,000 and $3,000 per unit), is relatively difficult to transport and setup and requires a relatively high level of technical expertise. To address these issues, Performance Edge VR has now been converted for delivery on a stand-alone, inside-out tracking Oculus Quest headset with a simple, single-strap respiratory belt (Vernier, USA) that connects directly to the headset via USB-C cable. This new system significantly reduces overall costs (<AU$1,000 per unit), is easy to transport, store and set-up and can be used in parallel by multiple trainees in the same space, therefore improving usability in a real-world training setting.

There is now a need to extend structured studies to inform the implementation approach and document training outcomes in additional participants, particularly within the target population (i.e. military trainees). Although our data provides preliminary evidence that Performance Edge VR can provide effective breath control training it does not validate its effectiveness as a stress management skill *per se*. Further research is planned to assess

Performance Edge VR training in a stress exposure context, including potential inclusion of VR-based stressor scenarios. Ongoing research should incorporate a controlled study design to support comparison with a control group (e.g. generic, non-VR stress management training) and assess the long-term retention of training outcomes. It would also be useful to validate future Performance Edge VR versions in multiple user-groups, ranging from law enforcement, first responder and paramedics to the general public or mental health patients (e.g. anxiety sufferers).

## Supporting information

**S1 File.**
(DOCX)

## Acknowledgments

We thank Centre for Advanced Training Systems members Ann Stevenson, Angela Keynes and Rebecca Hood for constructive feedback and ideas during development and internal testing of Performance Edge VR. We thank external consultants Liz Ditton and Brendon Knott for providing their expertise on CBT theory to inform the development of scripts and practical exercises. We appreciate the efforts of Australian Defence Force members in initiation and feedback on this project, as well as supporting ongoing implementation and assessment in a military training setting. Finally, we thank Jumpgate VR (Edward Thomas, Lewis McLauchlan and Anton Andreacchio) for their work and input coding Performance Edge VR.

## Author Contributions

**Conceptualization:** Murielle G. Kluge, Nicole Walker, Eugene Nalivaiko, Frederick Rohan Walker.

**Formal analysis:** Murielle G. Kluge, Steven Maltby.

**Funding acquisition:** Nicole Walker, Neanne Bennett, Eugene Aidman, Eugene Nalivaiko, Frederick Rohan Walker.

**Investigation:** Murielle G. Kluge, Steven Maltby, Nicole Walker, Neanne Bennett, Eugene Nalivaiko, Frederick Rohan Walker.

**Methodology:** Murielle G. Kluge, Eugene Nalivaiko, Frederick Rohan Walker.

**Project administration:** Eugene Nalivaiko, Frederick Rohan Walker.

**Resources:** Nicole Walker, Neanne Bennett, Eugene Aidman, Eugene Nalivaiko, Frederick Rohan Walker.

**Supervision:** Eugene Nalivaiko, Frederick Rohan Walker.

**Writing – original draft:** Murielle G. Kluge, Steven Maltby, Eugene Nalivaiko, Frederick Rohan Walker.

**Writing – review & editing:** Murielle G. Kluge, Steven Maltby, Nicole Walker, Neanne Bennett, Eugene Aidman, Eugene Nalivaiko, Frederick Rohan Walker.

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
