## [Decision Letter · Decision Letter 0]

1 Dec 2020

PONE-D-20-29906

Development of a modular stress management platform (Performance Edge VR) and a pilot efficacy trial of a bio-feedback enhanced training module for controlled breathing

PLOS ONE

Dear Dr. Maltby,

Thank you for submitting your manuscript to PLOS ONE. After careful consideration, we feel that it has merit but does not fully meet PLOS ONE’s publication criteria as it currently stands. Therefore, we invite you to submit a revised version of the manuscript that addresses the points raised during the review process.

We look forward to receiving your revised manuscript.

Kind regards,

Bijan Najafi

Academic Editor

PLOS ONE

Journal Requirements:

3.Thank you for stating the following in the Competing Interests section:

[All authors were involved in the development of the Performance Edge VR application. Intellectual property relating to the application is owned by The University of Newcastle. The Australian Defence Innovation Hub provided funding for application development and retains rights for its future use.].

4.In your Methods section, please provide additional information about the participant recruitment method and the demographic details of your participants. Please ensure you have provided sufficient details to replicate the analyses such as:

a) the recruitment date range (month and year),

b) a description of any inclusion/exclusion criteria that were applied to participant recruitment,

c) a table of relevant demographic details,

d) a statement as to whether your sample can be considered representative of a larger population.

5. Please include additional information regarding the post-training questionnaires used in the study and ensure that you have provided sufficient details that others could replicate the analyses. For instance, if you developed the questionnaire(s) as part of this study and it is not under a copyright more restrictive than CC-BY, please include a copy, in both the original language and English, as Supporting Information.

6. For papers that describe new methods or tools, these reports must meet the criteria of utility, validation, and availability, which are described in detail at http://journals.plos.org/plosone/s/submission-guidelines#loc-methods-software-databases-and-tools. Please describe how another researcher could access the Performance Edge VR training module, or provide a link to your code.

Reviewers' comments:

Reviewer's Responses to Questions

**Comments to the Author**

1. Is the manuscript technically sound, and do the data support the conclusions?

Reviewer #1: Yes

Reviewer #2: Yes

2. Has the statistical analysis been performed appropriately and rigorously? 

Reviewer #1: Yes

Reviewer #2: No

3. Have the authors made all data underlying the findings in their manuscript fully available?

Reviewer #1: No

Reviewer #2: Yes

4. Is the manuscript presented in an intelligible fashion and written in standard English?

Reviewer #1: Yes

Reviewer #2: Yes

5. Review Comments to the Author

Reviewer #1: The manuscript is very well-written and presented, however, I do have few comments/suggestions that may improve the readability.

Lack of line numbers make it hard to point out specific comments, but I will try to do my best.

Intro/Methods: Please also provide some relevant literature on importance of breathing variability for stress management before directly introducing it in Methods as one of the things you assessed from the VR module.

Overall, I like the discussion where you compare with (many) other similar platforms/devices out there trying to make use of breathing/HR/HRV for biofeedback for stress management, however, I think some of that could be moved up to the Introduction / methods to help make the case as why you chose to do the way you chose to do it.

Overall, it seems there is over abundance of re-emphasizing as how the design went through multiple iterations and got approval from so and so, but I think some of that is internal to your logistics and unnecessary distraction. It will be great to have more focus on technical aspects of the design. Remember although PLoS one covers quite broad topics, but it is not a typical HCI journal.

2.7 Study protocol: I think it will help to provide more technical details about the type/order/cutoff of filter being used. Did you use all 3 min. of data, or removed first 30-45 sec (I think the manual for Zephyr Bioharness suggest that it may take up to 45 sec to reach a stable level after start of any condition).

I think tracking depth of breadth is very important and if the raw data has been collected w/o real-time digital filtering, then it should be discussed, even if authors do not currently use it to provide feedback (but can comment why this may not also be used). I think for a within-subject design, it may be okay to use this metric even after low pass filtering, to help explain if they are engaging in more deeper breaths. Were subjects told to do diaphragmatic breathing or no such instructions were given, if not, why that was not deemed necessary for a breathing intervention? Could it not make some people who may have bias about breathing control from linking it to meditation/yoga to not try to think of this like that?

Section 3.2.....

Please provide a higher resolution snapshot in Fig. 2. It is not legible (else I do not see any point of providing it)

When subject was given bio-feedback, was that of the BR in real-time, or average of last few seconds (if not, why should some sort of moving average filter not be considered?)

I see Fig. 2D says 4 bpm that is really low to be considered smooth/continuous/repeatable/sustainable breathing. Please discuss implications for such very low breathing (and thus low O2?).

Similarly what are the three numbers 20, 8 and 34 in Fig. 2C?

Were subjects told to have some target breathing? If so, how was that target determined or the trajectory that they were supposed to follow? How were they scored / performance assessed?

.......

3.4 How long time interval was used to estimate 'lowest comfortable breathing rate'?

pg 12 top line - where you say at first they used the live respiratory trace as a visual guide - I wonder how do they know if it is not correct or what is that they should try to do to change it to what?

4 lines down when you say performance is provided to the user, please provide more information as how they were scored?

Please provide error bars in Figs 3 and 4 and not just averages across 30 subjects for each of the categories.

It was also not clear if some stressors can be integrated in this platform to test this training module for its efficacy as a stress management tool per se.

Reviewer #2: “In fact, there is now growing recognition that optimal performance in challenging situations (and importantly during post-exposure recovery), is determined by a positive interaction between external and internal skills and knowledge (9).”

9. Zheng C, Kashi K, Fan D, Molineux J, Ee MS. Impact of individual coping strategies and organisational work–life balance programmes on Australian employee well-being. The International Journal of Human Resource Management. 2016;27(5):501-26.

This does not really seem enough to say “there is growing recognition…”

Please fix “‘inoculuate”

Figure 2 should be better quality in the final version because when I zoom on it I can’t really see what’s written.

In "3.1 Identification of a need for new training approaches to support stress management training in Defence"

These are supposed to be results. I’d like to see something showing a result, not just a sentence saying “it is necessary”

“Several potential participants opted not to participate after reading the participant information statement. “

I wonder how many, and whether the sample of people who wanted to do this is somehow biased to high acceptance.

“Half of all participants reported previous experience with controlled breathing at the beginning of session 1” Where these analyzed separate from others? This should be done to make sure previous experience is not biasing the results.

The data in figure 3 should somehow show deviation. Given the small sample size, it would be easy to show all results. Any other form of showing at least the range would be good.

Statistics for figures 3 and 5 would help.

I feel the manuscript could be shorter without losing its message.

6. PLOS authors have the option to publish the peer review history of their article (what does this mean?). If published, this will include your full peer review and any attached files.

Reviewer #1: No

Reviewer #2: No

---

## [Author Response · Author response to Decision Letter 0]

16 Dec 2020

A detailed point-by-point response to editor and reviewer feedback has been uploaded as a separate "Response to Reviewers" file.

DETAILED POINT-BY-POINT RESPONSE TO COMMENTS:

Editorial Office Feedback:

Article formatting has been updated to comply with PLOS ONE style requirements and guidelines.

The indicated sentence referred to measurements of baseline unprompted breathing rates at the beginning of each training session. This sentence has been removed from the manuscript, as the point is not critical to the key study findings.

[All authors were involved in the development of the Performance Edge VR application. Intellectual property relating to the application is owned by The University of Newcastle. The Australian Defence Innovation Hub provided funding for application development and retains rights for its future use.].

The Competing Interests statement has been updated (pg. 21). The updated statement is also included in the cover letter, as requested.

4. In your Methods section, please provide additional information about the participant recruitment method and the demographic details of your participants. Please ensure you have provided sufficient details to replicate the analyses such as:

a) the recruitment date range (month and year),

b) a description of any inclusion/exclusion criteria that were applied to participant recruitment,

c) a table of relevant demographic details,

d) a statement as to whether your sample can be considered representative of a larger population.

We have included additional detail on the study recruitment dates and inclusion / exclusion criteria and representativeness of the study population in the Methods (pg. 7/8). Relevant demographic details are included in Table I.

5. Please include additional information regarding the post-training questionnaires used in the study and ensure that you have provided sufficient details that others could replicate the analyses. For instance, if you developed the questionnaire(s) as part of this study and it is not under a copyright more restrictive than CC-BY, please include a copy, in both the original language and English, as Supporting Information.

Questionnaires were developed for the current study and have now been included in Supporting Information, as noted in the Methods (pg. 9)

6. For papers that describe new methods or tools, these reports must meet the criteria of utility, validation, and availability, which are described in detail at http://journals.plos.org/plosone/s/submission-guidelines#loc-methods-software-databases-and-tools. Please describe how another researcher could access the Performance Edge VR training module, or provide a link to your code.

We have added the following statement in the Methods section (pg. 6/7) – “Intellectual property relating to Performance Edge VR is owned by The University of Newcastle. If interested in accessing Performance Edge VR application for research purposes, please contact the corresponding author.”

Review Comments to the Author:

Reviewer #1: 

The manuscript is very well-written and presented, however, I do have few comments/suggestions that may improve the readability. 

We thank the reviewer for their positive and constructive feedback.

Intro/Methods: Please also provide some relevant literature on importance of breathing variability for stress management before directly introducing it in Methods as one of the things you assessed from the VR module.

A sentence has been added to the introduction clarifying that controlled breathing consists of deep, slow and steady breathing which can be measured as a reduction in both respiratory rate and variability (p.4). This is outlined in references 21, 22. 

Overall, I like the discussion where you compare with (many) other similar platforms/devices out there trying to make use of breathing/HR/HRV for biofeedback for stress management, however, I think some of that could be moved up to the Introduction / methods to help make the case as why you chose to do the way you chose to do it.

As suggested, we have included a brief summary of existing VR applications integrating biofeedback in the Introduction (pg. 5).

Overall, it seems there is over abundance of re-emphasizing as how the design went through multiple iterations and got approval from so and so, but I think some of that is internal to your logistics and unnecessary distraction. It will be great to have more focus on technical aspects of the design. Remember although PLoS one covers quite broad topics, but it is not a typical HCI journal.

Efforts have been made to streamline the indicated Methods & Results sections, where we describe the scoping, approval and development processes. The remaining content related to review and approval was deemed relevant to highlight the efforts made to ensure that Performance Edge VR is fit-for-purpose and suited to the intended training environment, and that it has been supported at the highest organisational level of the Australian Defence Forces.

2.7 Study protocol: I think it will help to provide more technical details about the type/order/cutoff of filter being used. Did you use all 3 min. of data, or removed first 30-45 sec (I think the manual for Zephyr Bioharness suggest that it may take up to 45 sec to reach a stable level after start of any condition). I think tracking depth of breadth is very important and if the raw data has been collected w/o real-time digital filtering, then it should be discussed, even if authors do not currently use it to provide feedback (but can comment why this may not also be used). I think for a within-subject design, it may be okay to use this metric even after low pass filtering, to help explain if they are engaging in more deeper breaths. Were subjects told to do diaphragmatic breathing or no such instructions were given, if not, why that was not deemed necessary for a breathing intervention? Could it not make some people who may have bias about breathing control from linking it to meditation/yoga to not try to think of this like that?

Thank you for this comment. We have added more technical details to the methods section (p. 8). The Equivital life monitor belt used in this study provides real-time data capture and does not have a time delay as suggested for the Zephyr Bioharness. Pre- and post-training breathing data was collected for 3-4min, whilst 3 consecutive minutes of the recording was analysed. A digital Triangular (Bartlett) Smoothing window with window width of 205 samples was used to filter real-time raw data, necessary to remove artefacts and instrumental noise. This has now been noted in the manuscript Methods (pg. 8) Whilst we agree with the reviewer’s comment stating that breathing depth is a relevant component of breath control, no analysis was performed due to effects of smoothing on this output. Instead, we focused on breathing rate and consistency as valid measures of breath control as they were not affected by data processing. 

Participants were instructed to take deep, slow and steady breaths at their personal lowest comfortable breathing rate with no specific reference to diaphragmatic breathing or any other breathing technique. We prioritised easy to follow instructions over a specific rhythm or technique with the learning goal focused on the development of breath awareness for intentional and controlled breathing. This decision is supported by existing literature, which indicates that many different breathing techniques can be effective in the context of stress management, provided they include intentional and rhythmical breathing at a reduced breathing rate (Discussion; p.17-18, line 519-559). Increased breathing depth is a feature which often accompanies intentional, rhythmical breathing at a reduced rate but was not used for biofeedback within our VR application or a primary outcome of this study due to the above-mentioned filtering of the real-time data, which was required for the live biofeedback tool. 

Section 3.2 - Please provide a higher resolution snapshot in Fig. 2. It is not legible (else I do not see any point of providing it)

A higher resolution image has been uploaded for Figure 2. 

When subject was given bio-feedback, was that of the BR in real-time, or average of last few seconds (if not, why should some sort of moving average filter not be considered?)

Users were provided with a real-time respiratory bio-feedback trace and a moving average breathing rate, when exercises integrated live biofeedback. In other exercises, users were provided with their average breathing rate over a 2-minute timeframe and a breathing trace, after completing the exercise. This information is represented in Figure 2 and has been clarified in the figure legend (pg. 10).

I see Fig. 2D says 4 bpm that is really low to be considered smooth/continuous/repeatable/sustainable breathing. Please discuss implications for such very low breathing (and thus low O2?).

The indicated figure was provided to visualise the graphic design and user interface for Performance Edge VR. The respiratory data included in the figures shows the respiratory trace and rate of one of the research team members, and was not intended to be representative of usage data. An updated Figure 2 has now been included.

We note that users were prompted to reduce their breathing rates to their personal “lowest comfortable breathing rate”. The lowest average breathing rate recorded for the study cohort after training session 3 was approximately 4.9 BPM (Figure 6A). No measurements were taken to assess blood oxygenation levels. We do note that literature suggests breathing rates as low as 3 BPM can be performed by experts without effecting oxygen levels (due to associated increases in depth of breathing). 

Similarly what are the three numbers 20, 8 and 34 in Fig. 2C?

The screenshot displays the user’s average breathing rate at baseline (“Normal”; top), after completing 10 deep breaths (middle) and after mild physical activity (standing & sitting 10 times; bottom) to familiarise the user with the physiological changes in their breathing rate. A higher resolution screenshot has been included and the figure legend has been modified to clarify this point (Figure 2; pg. 10).

Were subjects told to have some target breathing? If so, how was that target determined or the trajectory that they were supposed to follow? How were they scored / performance assessed?

A common feature across effective breathing approaches is intentional and rhythmical breathing at a reduced breathing rate. In alignment with this, Performance Edge VR does not impose a specific target value but rather provides training to maintain a slow and steady breath at a personal user-defined “lowest comfortable breathing rate”. Biofeedback is used in Performance Edge VR as an educational training feature with 3 distinct goals: 1) Visualisation of changes in personal breathing pattern and rates for different activities (Figure 2C), 2) Visualisation of real-time breathing trace to support manual control over breathing and assist with establishing a lowest comfortable breathing rate (depicted in Figure 2B) and as a 3) Performance measure of the trainees ability to maintain their personal lowest comfortable breathing rate in subsequent exercises (without the assistance of real-time biofeedback and with accompanying distractions). Feedback in each exercise was reported to users in comparison to their own personal lowest comfortable breathing rate (ie. faster, same or slower). This was presented after each exercise to provide feedback to users (Figure 2D). Figure 2, the accompanying figure legend and manuscript text (p. 11/12) have been modified to clarify these points. Of note, in the final exercise users perform a secondary shooting task whilst maintaining their lowest comfortable breathing rate. In this case, “performance” feedback included details on both maintenance of breathing rate AND shooting game outcomes (ie. total time, number of arrows used and points to provide some gamification of relative performance). 

3.4 How long time interval was used to estimate 'lowest comfortable breathing rate'?

Within the Performance Edge VR application, the lowest comfortable breathing rate was measured over a 2-minute period during exercise 2. This measurement is used to provide user feedback for comparison in subsequent exercises, as described in detail above. In the research trial protocol, controlled breathing was assessed both pre- and post-training during a 3-minute time period (Figure 1). In both circumstances, users were prompted to “take slow and steady breaths at their lowest comfortable breathing rate”. This is indicated in the Methods (pg. 8).

pg 12 top line - where you say at first they used the live respiratory trace as a visual guide - I wonder how do they know if it is not correct or what is that they should try to do to change it to what?

4 lines down when you say performance is provided to the user, please provide more information as how they were scored?

Clarification has been integrated into the manuscripts as outlined in the responses to reviewer comments above. 

Please provide error bars in Figs 3 and 4 and not just averages across 30 subjects for each of the categories.

Figures 3 & 4 have been updated to include errors bars or individual user data points, as requested. 

It was also not clear if some stressors can be integrated in this platform to test this training module for its efficacy as a stress management tool per se.

Thank you for this suggestion. Indeed, we intend to add stressors in future expansions of Performance Edge VR and we agree that this will be a useful application of the technology. This has now been noted in the manuscript Discussion (pg. 20). 

Reviewer #2: 

“In fact, there is now growing recognition that optimal performance in challenging situations (and importantly during post-exposure recovery), is determined by a positive interaction between external and internal skills and knowledge (9).”

9. Zheng C, Kashi K, Fan D, Molineux J, Ee MS. Impact of individual coping strategies and organisational work–life balance programmes on Australian employee well-being. The International Journal of Human Resource Management. 2016;27(5):501-26.

This does not really seem enough to say “there is growing recognition…”

The indicated sentence has been removed.

Please fix “‘inoculuate”

Typo corrected (pg. 5).

Figure 2 should be better quality in the final version because when I zoom on it I can’t really see what’s written.

An updated high resolution figure has been uploaded. 

In "3.1 Identification of a need for new training approaches to support stress management training in Defence"

These are supposed to be results. I’d like to see something showing a result, not just a sentence saying “it is necessary”

The content identified has now been moved into the Methods section (pg. 6) to provide context for project initiation. Specific qualitative / quantitative data was not collected for this aspect of the study. 

“Several potential participants opted not to participate after reading the participant information statement. “

I wonder how many, and whether the sample of people who wanted to do this is somehow biased to high acceptance.

We have updated the text to clarify that 3 potential participants opted not to participate in the study after reading the participant information document, due to the length of the study and requirement to attend 3 training visits (pg. 12). We do not believe this factor is likely to have biased the findings or affected the application to the broader population.

“Half of all participants reported previous experience with controlled breathing at the beginning of session 1” Where these analyzed separate from others? This should be done to make sure previous experience is not biasing the results.

Thank you for this suggestion and great point. We have now performed a post-hoc comparison of participants with vs. those without previous experience in controlled breathing and included the data as a new Figure 7 (pg. 15). Participants with previous experience in controlled breathing had reduced objective breathing metrics compared to those without experience at training session 1. However, both groups of participants had a reduction in breathing rate and variability after training (pre vs. post). Despite this pilot study not being statistically powered for comparisons of users based on previous experience, this dataset suggests that Performance Edge VR can improve objective breathing metrics for both novices and trainees with experience in controlled breathing. As noted, no details were collected in the current study about the relative levels of pre-existing user experience, and this would be useful to capture in future research (pg. 16).

The data in figure 3 should somehow show deviation. Given the small sample size, it would be easy to show all results. Any other form of showing at least the range would be good.

Figures 3 has been updated to include errors bars, which has now been noted in the figure legend. 

Statistics for figures 3 and 5 would help.

We believe the reviewer is referring to Figures 3 & 4. No statistical analysis was performed or reported for these datasets, as comparisons of self-reported data was not considered relevant study outcomes and the study was not powered to assess these aspects. 

I feel the manuscript could be shorter without losing its message.

Efforts have been made to streamline the manuscript, while maintaining the key points and messages. Aspects of the Methods and Results relating to project scoping, approvals and initial design decisions were significantly shortened, and the overall revised manuscript length is reduced despite addition of content to address reviewers’ comments as outlined above (previous submission = 7268 words in main text; updated version = 7029 words)

---

## [Decision Letter · Decision Letter 1]

22 Dec 2020

Development of a modular stress management platform (Performance Edge VR) and a pilot efficacy trial of a bio-feedback enhanced training module for controlled breathing

PONE-D-20-29906R1

Dear Dr. Maltby,

We’re pleased to inform you that your manuscript has been judged scientifically suitable for publication and will be formally accepted for publication once it meets all outstanding technical requirements.

Kind regards,

Bijan Najafi

Academic Editor

PLOS ONE

Additional Editor Comments (optional):

Thanks for addressing reviewers' voiced concerns. After reviewing your revision and the response letter, I believe the revision is responsive to initial concerns and manuscript has sufficient scientific merit. I however noticed that you used a URL in the manuscript manuscript text, line 142. Please remove the URL to the citation section and add the latest access date. This could be however done during Proofs reading process and I don't want to delay acceptance of your manuscript with this minor issue. Thus I am happy to recommend acceptance of your manuscript as it is. Congratulation!

Reviewers' comments:

Reviewer's Responses to Questions

**Comments to the Author**

1. If the authors have adequately addressed your comments raised in a previous round of review and you feel that this manuscript is now acceptable for publication, you may indicate that here to bypass the “Comments to the Author” section, enter your conflict of interest statement in the “Confidential to Editor” section, and submit your "Accept" recommendation.

Reviewer #1: All comments have been addressed

2. Is the manuscript technically sound, and do the data support the conclusions?

Reviewer #1: Yes

3. Has the statistical analysis been performed appropriately and rigorously? 

Reviewer #1: Yes

4. Have the authors made all data underlying the findings in their manuscript fully available?

Reviewer #1: Yes

5. Is the manuscript presented in an intelligible fashion and written in standard English?

Reviewer #1: Yes

6. Review Comments to the Author

Reviewer #1: I am satisfied with authors' responses, editing and additions. I recommend this manuscript be accepted.

7. PLOS authors have the option to publish the peer review history of their article (what does this mean?). If published, this will include your full peer review and any attached files.

Reviewer #1: **Yes: **Rahul Goel

---

## [Editor Report · Acceptance letter]

20 Jan 2021

PONE-D-20-29906R1 

Development of a modular stress management platform (Performance Edge VR) and a pilot efficacy trial of a bio-feedback enhanced training module for controlled breathing 

Dear Dr. Walker:

I'm pleased to inform you that your manuscript has been deemed suitable for publication in PLOS ONE. Congratulations! Your manuscript is now with our production department. 

Kind regards, 

on behalf of

Dr. Bijan Najafi 

Academic Editor

PLOS ONE